# Hippocampal subfield volumes in abstinent men and women with a history of alcohol use disorder

Kayle S. Sawyer[1,2,3,4]⊛*, Noor Adra[1,3]⊛, Daniel M. Salz[1,2,3]⊛, Maaria I. Kemppainen[1,2,3], Susan M. Ruiz[1,2,3], Gordon J. Harris[2,3,5], Marlene Oscar-Berman[1,2,3]

**1** VA Boston Healthcare System, Boston, MA, United States of America, **2** Boston University School of Medicine, Boston, MA, United States of America, **3** Massachusetts General Hospital, Boston, MA, United States of America, **4** Sawyer Scientific, LLC, Boston, MA, United States of America, **5** Harvard Medical School, Boston, MA, United States of America

⊛ These authors contributed equally to this work.
* kslays@bu.edu

**Data Availability Statement:** Data and code are available at https://gitlab.com/kslays/moblab-hippocampus.

## Abstract

Alcohol Use Disorder (AUD) has been associated with abnormalities in hippocampal volumes, but these relationships have not been fully explored with respect to sub-regional volumes, nor in association with individual characteristics such as age, gender differences, drinking history, and memory. The present study examined the impact of those variables in relation to hippocampal subfield volumes in abstinent men and women with a history of AUD. Using Magnetic Resonance Imaging at 3 Tesla, we obtained brain images from 67 participants with AUD (31 women) and 64 nonalcoholic control (NC) participants (31 women). The average duration of the most recent period of sobriety for AUD participants was 7.1 years. We used Freesurfer 6.0 to segment the hippocampus into 12 regions. These were imputed into statistical models to examine the relationships of brain volume with AUD group, age, gender, memory, and drinking history. Interactions with gender and age were of particular interest. Compared to the NC group, the AUD group had approximately 5% smaller subiculum, CA1, molecular layer, and hippocampal tail regions. Age was negatively associated with volumes for the AUD group in the subiculum and the hippocampal tail, but no significant interactions with gender were identified. The relationships for delayed and immediate memory with hippocampal tail volume differed for AUD and NC groups: Higher scores on tests of immediate and delayed memory were associated with smaller volumes in the AUD group, but larger volumes in the NC group. Length of sobriety was associated with decreasing CA1 volume in women (0.19% per year) and increasing volume size in men (0.38% per year). The course of abstinence on CA1 volume differed for men and women, and the differential relationships of subfield volumes to age and memory could indicate a distinction in the impact of AUD on functions of the hippocampal tail. These findings confirm and extend evidence that AUD, age, gender, memory, and abstinence differentially impact volumes of component parts of the hippocampus.

**Funding:** This work was supported by funds from the National Institute on Alcohol Abuse and Alcoholism (NIAAA; https://www.niaaa.nih.gov/) of the National Institutes of Health US Department of Health and Human Services under Award Numbers R01AA07112 and K05AA00219 awarded to M.O.B.; US Department of Veterans Affairs Clinical Science Research and Development (https://www.research.va.gov/services/csrd/) grant I01CX000326 awarded to M.O.B; National Center for Advancing Translational Sciences of the National Institutes of Health US Department of Health and Human Services under Award Numbers 1S10RR023401, 1S10RR019307, 1S10RR023043, and 1UL1TR001430. The funders provided support in the form of salaries for authors K.S.S., D.M.S., M.I.K., S.M.R., G.J.H., and M.O.B., but did not have any additional role in the study design, data collection and analysis, decision to publish, or preparation of the manuscript. K.S.S. is an employee of Sawyer Scientific, LLC, and this affiliation provided no funding related to the work described in this manuscript. The specific roles of these authors are articulated in the 'author contributions' section. The content is solely the responsibility of the authors and does not necessarily represent the official views of the National Institutes of Health, the U.S. Department of Veterans Affairs, or the United States Government.

**Competing interests:** All authors declare that they have no competing financial interests in relation to the work described. K.S.S. is an employee of Sawyer Scientific, LLC, and this does not alter our adherence to PLOS ONE policies on sharing data and materials.

## Introduction

Magnetic resonance imaging (MRI) has been used extensively to study morphological changes in the brain associated with alcohol use disorder (AUD), a widespread and harmful condition [1,2]. Because memory impairments are associated with long-term chronic AUD, one neuroanatomical focus of investigation has been the hippocampus [3]. Not only has the hippocampus been shown to display the largest volume loss of seven subcortical structures examined in association with chronic AUD [4], a meta-analysis [5] summarized studies showing smaller volumes in alcohol using groups than in groups with no or minimal alcohol use. The hippocampus is a heterogeneous structure, with specific functions processed through overlapping internal networks, so analyses of subfields could help specify which functional networks are involved in AUD.

In identifying subfields responsible for functional abnormalities in networks associated with AUD, we considered the processing streams that exist within the hippocampus (Fig 1, which overlays this processing stream on an MRI atlas [6]). Traditionally, based upon anatomical research, hippocampal subfields have been defined as the subiculum, dentate gyrus, and *cornu ammonis* regions (CA1 through CA3) [7]. Researchers have focused on two major neural pathways, as follows: The first major pathway originates in entorhinal cortex, which transmits signals to the hippocampus. The pathway then proceeds along a trisynaptic circuit to (1) the dentate gyrus, then to (2) CA2 or CA3, and finally to (3) CA1 and subiculum. (Although the subiculum is not explicitly part of the hippocampus, it serves as an output.) The second major pathway also originates in entorhinal cortex, and it has a direct connection to CA1, among other regions. Both the direct and the trisynaptic pathways exist in parallel throughout the anterior, middle, and posterior hippocampus. That is, the slice shown in Fig 1B would look similar were the slice taken from a more anterior or from a more posterior section through the hippocampus. All of the input connections to CA1, CA2, CA3, and subiculum are made within an area called the molecular layer (also known as *stratum lacunosum moleculare* or SLM), and the fimbria consists of white matter fibers that carry projections from the hippocampus in a temporal to dorsal direction [8].

The direct pathway from entorhinal cortex is theorized to contain information regarding presently experienced stimuli [9,10]. For the structures involved in the trisynaptic pathway, the dentate gyrus has been shown to play a role in distinguishing different contexts [11], and the CA2 and CA3 regions have been ascribed to learning, memory encoding [12], early retrieval of verbal information [13], and disambiguation and encoding of overlapping representations [14]. The output fields, CA1 and subiculum, are thought to compare current context with remembered contexts [15]. Their projections include regions implicated in addictive disorders: the prefrontal cortex, amygdala, nucleus accumbens, and ventral tegmental area [16]. The anterior portion of the entire hippocampus (analogous to *ventral* in the rodent) has been considered to be involved in a wide variety of contexts including stress, emotion, and affect [17], while the posterior portion (analogous to *dorsal* in the rodent) has been demonstrated to have involvement in spatial processing [18].

Using software that provides automatic segmentation of hippocampal subfield volumes [6], three studies of abnormalities in AUD have reported smaller volumes of the subiculum, presubiculum, CA1, CA2+3, and CA4, or other parts of the hippocampus including dentate gyrus, hippocampal-amygdaloid transition area (HATA), and the fimbria [19–21]. In addition to brain abnormalities associated with AUD, interactions with age and gender have been revealed [22–32]. While the influence of age has been exemplified by a significant interaction of reduced volume of the CA2+3 region [19], hippocampal subfield projects in which women were included did not examine gender interactions [19,21]. Indeed, most studies of total

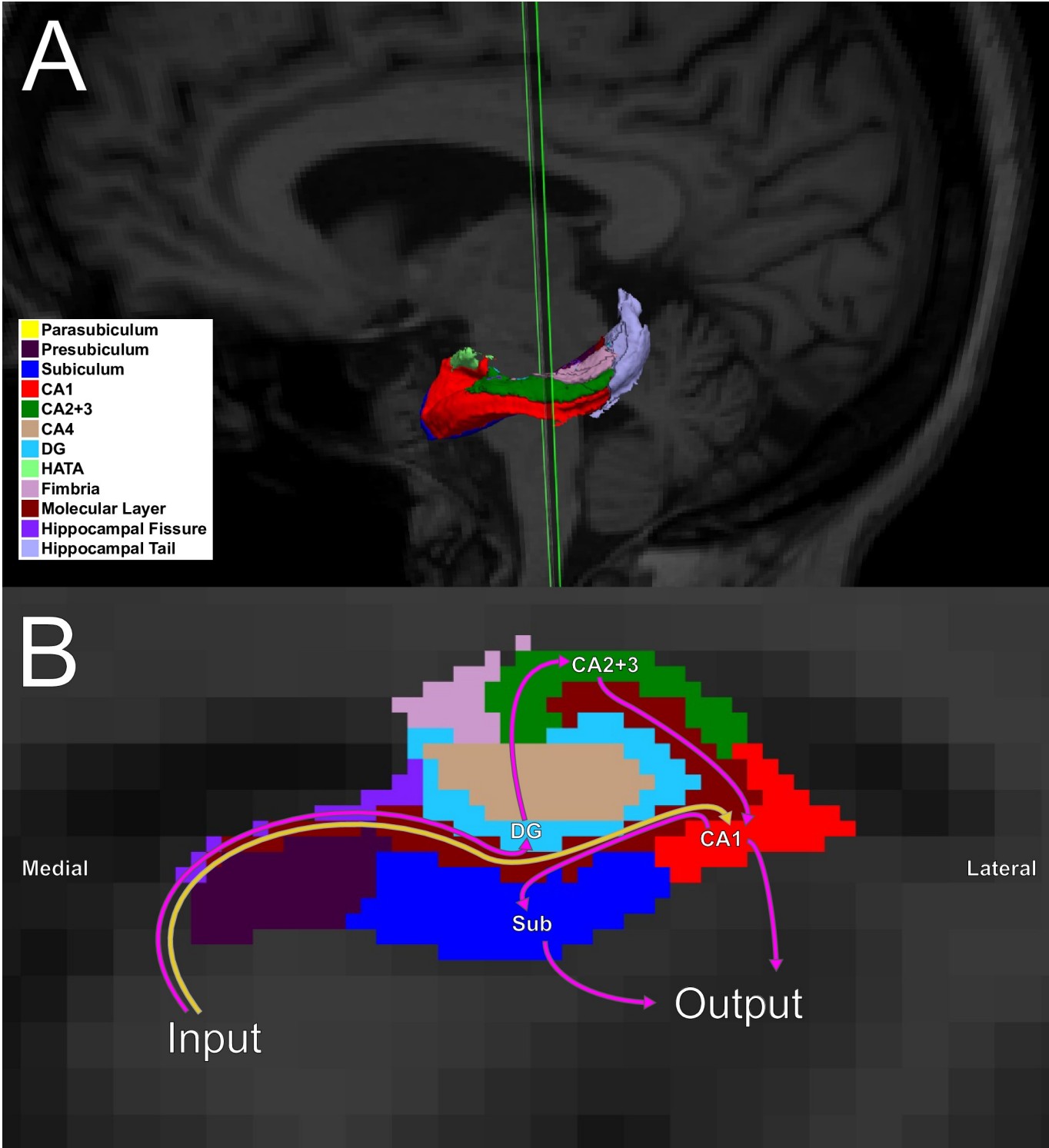

**Fig 1. Hippocampal subfield segmentation using the procedure by Iglesias and colleagues [6].** A. This three-dimensional model of the left hippocampus depicts its segmentation into 12 subfields (although not all subfields are visible from the angle presented). B. This coronal section through the hippocampus illustrates subfields that are visible in the slice, and it shows the two major hippocampal neural pathways originating in the entorhinal cortex and as described in the text: The yellow arrow shows the direct circuit; the pink arrows indicate the trisynaptic circuit. Although the connection from CA1 to the subiculum is not explicitly part of the trisynaptic circuit, it is important as an output region, in addition to CA1. Abbreviations: CA1 = cornu ammonis 1; CA2+3 = cornu ammonis 2 and 3; CA4 = cornu ammonis 4; DG = dentate gyrus; HATA = hippocampal-amygdaloid transition area; Sub = subiculum.

hippocampal volume in AUD-related abnormalities included men only [20,33–36], or did not examine gender interactions [19,21,37]. In studies that did analyze men with AUD (AUDm) and women with AUD (AUDw) separately, investigators observed lower hippocampal volumes in the AUDm and in the AUDw, but they did not find significant interactions between gender and diagnostic group [37], while we previously observed smaller hippocampal volumes for AUDm than AUDw [25].

Besides age and gender, two other factors are especially relevant when examining relationships of AUD with hippocampal subfield volumes: memory ability and drinking history. In neither the Lee et al. [20] nor the Kühn et al. [21] studies was memory assessed, and Zahr et al. [19] did not identify significant memory impairment associated with reduced hippocampal subfield volumes. This is surprising, considering that lesions to the hippocampus result in memory deficits [38–40]. Decreases in hippocampal gray matter volume in AUDm have been associated with executive functioning deficits [34] but another study did not find an association between anterior hippocampal volume and memory impairment [41]. These inconsistent findings might be explained in part due to differences in drinking patterns, especially the duration of abstinence. Kühn et al. [21] found significant normalization of volume in subfield CA2 +3 two weeks following withdrawal, and a significant negative association with the number of years drinking.

In summary, because the evidence regarding AUD-related hippocampal subfield dysmorphology is inconclusive, we sought to clarify the volumetric and functional abnormalities and their associations with age, gender, memory, and alcohol consumption characteristics. We examined the interaction of group-by-gender-by-region to assess how the impact of AUD differed for men and women. Also, we examined the interaction of group-by-gender-by-age to assess abnormal aging for men as compared to women. Finally, we investigated interactions with memory and drinking characteristics.

## Methods

As described in detail below, 67 AUD (31 women) and 64 NC participants (31 women) were included in analyses (see Results). Participants' memory, and drinking history were assessed. T1-weighted 3T MRI scans were obtained with 1x1x1.5 mm voxels. The 12 hippocampal subfield volumes were obtained with FreeSurfer 6.0. In the mixed models, *volume* was entered as the dependent variable, and *region*, *group*, *age*, *and gender* were the independent factors, with group-by-gender-by-region and group-by-gender-by-age interactions included. Following significant effects, additional models were constructed to examine relationships with memory measures, and with drinking history.

### Participants

The study originally included 146 participants: 73 abstinent adults with a history of chronic AUD (33 AUDw) and 73 adult controls without AUD (34 NCw). All were recruited locally through online and print advertisements. Participants provided written informed consent for participation in the study, which was approved by the Institutional Review Boards at the Boston VA Healthcare System, Massachusetts General Hospital, and Boston University School of Medicine. Exclusion criteria for participants included left-handedness, Korsakoff's syndrome, HIV, head injury with loss of consciousness greater than 15 minutes, stroke, seizures unrelated to AUD, schizophrenia, Hamilton Rating Scale for Depression (HRSD) [42] score over 14, and illicit drug use (except marijuana) greater than once a week within the past five years. Fifteen individuals were excluded from the study for the following reasons: Three AUD participants (1 woman) were excluded for illicit drug use, and another three (1 woman) for brain lesions or

head trauma. Six NCm were excluded for binge drinking, head trauma, or unusable scan data. Two NCw were excluded for claustrophobia or brain lesions, and one NCw was identified as an outlier, with brain hippocampal region volumes +/- 4 standard deviations. The final sample of 131 participants comprised a total of 67 AUD (31 women) and 64 NC participants (31 women) included in data analyses (Table 1).

Diagnostic criteria for study exclusion were based on a medical history interview, the HRSD, and a computer-assisted, shortened version of the Computerized Diagnostic Interview Schedule (DIS) [43]. The DIS provides diagnoses of lifetime psychiatric illnesses as defined by criteria established by the American Psychiatric Association. Full-Scale IQ scores and memory performance were measured through the Wechsler Adult Intelligence Scale (WAIS) and the Wechsler Memory Scale (WMS) [44], conducted by trained researchers. Incomplete WMS scores were obtained from one NCm and two AUDm and were excluded from the memory analyses.

A number of participants were taking medications for a variety of conditions, had used drugs earlier than five years before enrollment, or had a potentially confounding medical history. We included these participants with confounding factors, so that our sample would be more representative of the conditions present in the United States, thereby allowing for greater generalizability of the results. However, the presence of confounding factors in the sample may limit the interpretability of our findings. Therefore, in the analysis of the results, a subsample of 79 participants (14 AUDm, 12 AUDw, 27 NCm, 26NCw) was created consisting of

**Table 1. Participant characteristics.**

|  | AUDw | (N = 31) |  |  | AUDm | (N = 36) |  |  |
|---|---|---|---|---|---|---|---|---|
| Measure | Mean | SD | Min | Max | Mean | SD | Min | Max |
| Age (years) | 54.4 | 12.0 | 31.0 | 76.0 | 51.1 | 10.8 | 28.0 | 77.0 |
| Education (years)[b] | 14.7 | 1.8 | 12.0 | 19.0 | 14.1 | 1.9 | 12.0 | 18.0 |
| Full Scale IQ[bb] | 103.1 | 14.5 | 69.0 | 132.0 | 104.1 | 16.6 | 75.0 | 142.0 |
| Immediate Memory | 107.3 | 17.3 | 56.0 | 135.0 | 102.6 | 14.9 | 73.0 | 132.0 |
| Delayed Memory | 110.2 | 16.9 | 62.0 | 135.0 | 104.8 | 14.1 | 81.0 | 133.0 |
| DHD (years)[a] | 13.7 | 5.6 | 5.0 | 25.0 | 17.5 | 9.1 | 5.0 | 37.0 |
| DD (oz ethanol/day) | 8.8 | 7.2 | 2.5 | 34.8 | 11.8 | 6.8 | 2.3 | 26.6 |
| LOS (years)[aa] | 10.9 | 11.9 | 0.1 | 36.1 | 3.9 | 6.2 | 0.1 | 27.4 |
|  | NCw | (N = 31) |  |  | NCm | (N = 33) |  |  |
| Measure | Mean | SD | Min | Max | Mean | SD | Min | Max |
| Age (years) | 53.6 | 15.8 | 28.0 | 78.0 | 50.5 | 12.3 | 27.0 | 82.0 |
| Education (years) | 15.3 | 2.6 | 12.0 | 20.0 | 15.4 | 2.5 | 10.0 | 20.0 |
| Full Scale IQ | 110.1 | 15.4 | 82.0 | 141.0 | 112.7 | 14.7 | 89.0 | 141.0 |
| Immediate Memory | 112.5 | 16.0 | 89.0 | 143.0 | 110.5 | 15.4 | 84.0 | 140.0 |
| Delayed Memory | 114.7 | 12.4 | 93.0 | 146.0 | 114.2 | 17.4 | 78.0 | 147.0 |
| DHD (years) | 0.0 | 0.0 | 0.0 | 0.0 | 0.0 | 0.0 | 0.0 | 0.0 |
| DD (oz ethanol/day) | 0.3 | 0.3 | 0.0 | 1.1 | 0.2 | 0.2 | 0.0 | 0.7 |
| LOS (years) | 2.1 | 7.3 | 0.0 | 31.9 | 2.1 | 7.9 | 0.0 | 40.2 |

Minimum (Min), Maximum (Max), Means, and standard deviations (SD) are provided for the participant characteristics of AUDw, AUDm, NCw, and NCm. Delayed Memory scores (DMI) were not obtained from one AUDm, and one NCm. Immediate Memory scores (IMI) were not obtained from one AUDm and one NCm. Abbreviations: AUDw = women with a history of Alcohol Use Disorder; AUDm = men with a history of Alcohol Use Disorder; NCw = women without a history of AUD; NCm = men without a history of AUD; Full Scale IQ = Wechsler Adult Intelligence Scale Full Scale IQ; Immediate Memory = Wechsler Memory Scale Immediate Memory Index; Delayed Memory = Wechsler Memory Scale Delayed Memory Index; DHD = Duration of Heavy Drinking; DD = Daily Drinks; LOS = Length of Sobriety. AUDw vs AUDm: [a]$p<0.05$, [aa]$p<0.01$; AUD vs NC: [b]$p<0.05$, [bb]$p<0.01$.

"unconfounded" participants who were not currently taking psychotropic medications, and reported never having used illicit drugs more than once a week. Additionally, that subsample was restricted to individuals for whom no source indicated hepatic disease, nor any of the following disorders: major depressive, bipolar I or II, schizoaffective, schizophreniform, or generalized anxiety. All statistical effects of group reported for ANOVAs (including group interactions) remained significant for this unconfounded subsample.

Drinking history was assessed using Duration of Heavy Drinking (DHD), i.e., years of consumption of 21 drinks or more per week, and Length of Sobriety (LOS), which measures abstinence duration in years. The amount, type, and frequency (ounces of ethanol per day, roughly corresponding to daily drinks; DD) of alcohol use was measured for the last six months during which the participant drank alcohol [45]. Criteria for AUD participants included at least five years of previous alcohol abuse or dependence, and a minimum of four weeks of abstinence prior to testing, which is important for obtaining stable levels of performance after ethanol and its metabolites have been eliminated from the body [46].

Two NCw and one NCm had no prior history of drinking, whereas the remaining NC participants drank occasionally. Compared to the AUDw, the AUDm had greater periods of heavy drinking and shorter periods of abstinence, which are consistent with national trends [47] and allow for generalizability of the results. However, to improve interpretability, we created a subsample in which the AUDw and AUDm were not significantly different by removing four AUDw with the longest LOS and shortest DHD values. All statistical effects regarding gender interactions with drinking history reported in this manuscript remained significant for this subsample.

## MRI acquisition and analysis

MRI scans were obtained at the Martinos Center for Biomedical Imaging at Massachusetts General Hospital on a 3 Tesla Siemens (Erlangen, Germany) MAGNETOM Trio Tim scanner with a 32-channel head coil. Image acquisitions included two T1-weighted multiecho MPRAGE scans collected for volumetric analysis that were averaged to aid in motion correction (TR = 2530 ms, TE = 1.79 ms, 3.71 ms, 5.63 ms, 7.55 ms [root mean square average used], flip angle = 7 degrees, field of view = 256 mm, matrix = 256 x 256, slice thickness = 1 mm with 50% distance factor, 176 interleaved sagittal slices, GRAPPA acceleration factor = 2; voxel size = 1.0 mm x 1.0 mm x 1.5 mm). T2 scans were unavailable for use in improving segmentation accuracy beyond the accuracy obtained with the T1 scans.

Scans were analyzed using an automated hippocampal segmentation method [6] in Free-Surfer (https://surfer.nmr.mgh.harvard.edu), which more recently has shown high reliability and agreement with manual segmentations [48]. Brain reconstructions were manually inspected and errors were corrected. Volumes of the hippocampal subfields (12 per hemisphere) were calculated using the hippocampal subfields subroutines for Freesurfer 6.0 (which omits the alveus due to the poor reliability of the segmentation). These 12 volumes (per hemisphere) were used to define the extent of the hippocampus. Estimated total intracranial volume was taken from the segmentation volume estimate [49].

## Statistical analyses

Statistical analyses were performed using R version 3.4.0 [50]. We used hierarchical linear models [51] to investigate the impact of several variables on regional hippocampal volumes. Data and code are available at https://gitlab.com/kslays/moblab-hippocampus. Because brain volumes vary with head size, we corrected for cranial volume, as follows: The NC volumes were fit to the estimated total intracranial volume (eTIV) values, and the slope 's' was obtained

from the NC group. Individual volumes for NC and AUD participants were adjusted using the formula V' = V − s × (eTIV − mean eTIV). Left and right hemisphere values of corresponding regions were averaged. In the regression models, we visually confirmed that other regression assumptions were satisfied (normality, homogeneity of variance, homogeneity of regression), and set thresholds for multicollinearity (Pearson correlations among predictors were < 0.5) and influence (Cook's D < 1.0).

We first conducted an analysis of total hippocampal volume using multiple regression. We constructed a model predicting whole hippocampal volume from the interaction of group, age, and gender, along with the lower-order interactions and main effects. Next, for our analysis of subfield volumes, we used mixed models. Volume was entered as the dependent variable, and group, region, age, and gender, were the independent factors (all fixed effects). In order to account for multiple observations per participant (i.e., volumes for each region), individual subject effects were specified as random intercepts.

Four statistical models were used for this project: (1) The primary model included the factors of group, age, gender, and subfield; secondary models additionally included (2) immediate memory scores, (3) delayed memory scores, and (4) drinking history measures. For each of the four models, we report findings from the ANOVA, followed by the results from the post hoc analyses (see Results). We examined the interaction of group-by-gender-by-region to assess how the impact of AUD differed for men and women, and how gender differences impacted certain regions in comparison to others. Also, we examined the interaction of group-by-gender-by-age to assess abnormal aging for men as compared to women. A four-way interaction of group-by-gender-by-region-by-age was used to confirm homogeneity of regression slopes. All non-significant (i.e., $p > 0.05$) interactions (except lower-order interactions included in higher-order interactions) that had been added to confirm homogeneity of regression slopes were removed, and the subsequent model was used to report results.

Following significant interactions, we evaluated the estimated marginal mean differences within each region using Bonferroni multiple comparisons correction of a family-wise $p$-value threshold of 0.05. Since there were 12 hippocampal subfields, this correction resulted in an adjusted $p$-value threshold of 0.0042. In order to reduce the number of subsequent tests (thereby reducing false positives) we limited our examination of post hoc analyses for the three secondary models (involving two memory measures and drinking history) to the subfields found to show significant group differences in the primary model. Post hoc results were reported as percent differences, using the following procedures: For group comparisons, each mean difference in subfield volume was divided by the mean subfield volume for the entire sample. For continuous relationships (age, memory measures, and drinking history), each slope was divided by the mean subfield volume for the entire sample. For interactions of group-by-gender, the comparisons of AUDm vs. NCm and AUDw vs. NCw were planned, while for region-by-group, only AUD vs. NC comparisons were planned. For age interactions, the effect of age for each of the subgroups was assessed, and the slope differences were compared in the same manner as mean differences.

To address the aims of our secondary models, we included the factors from our primary model. We investigated interactions of group-by-gender with two memory measures from the WMS: the Immediate Memory Index (IMI) and the Delayed Memory Index (DMI), using a separate analysis for each of the two index scores (group-by-gender-by-IMI in one model, and group-by-gender-by-DMI in the other). We then applied the same procedure used in the primary model to assess homogeneity of regression slopes (i.e., by examining group-by-gender-by-IMI-by-region and group-by-gender-by-DMI-by-region) and to remove extraneous interactions. Following significant interactions, we examined (a) the significance of each score's slope in the regions observed to be significant for the primary model, and (b) the same group

differences in slopes as for the primary model. In summary, the IMI and DMI scores were analyzed in separate ANOVA models, each of which contained between-subjects factors of group and gender, and a group-by-gender interaction term.

The fourth model investigated the relationship between the measure of subfield volume and measures of alcohol drinking history (DD, DHD, and LOS) among the AUD participants. The NC participants were not included because the participants had negligible variation within the scores. We examined interactions of each of these three measures with gender-by-region in a single model. As with the primary and the other secondary models, we tested for homogeneity of regression slopes, and removed interactions as dictated by significance levels. Following significant interactions, we examined the significance of each score's slope in the regions significant for the primary analyses, and we compared the slopes for men to the slopes for women.

## Results

### Participant characteristics

Table 1 provides information about the participants. The groups consisted of 67 AUD (31 women; 36 men) and 64 NC (31 women; 33 men), with a mean age of 52 years for the 131 participants. Between-group (NC vs. AUD) differences in age were not significant (95% CI of mean difference = [5.10, -3.82] years). The AUD participants had 0.98 fewer years of education ($t(117.18) = -2.51$, $p < 0.05$) and 7.82 points lower FSIQ scores ($t(129.00) = -2.93$, $p < 0.01$). The average duration of the most recent period of sobriety for AUD participants was 7.1 years. The AUDw had 7.00 years longer LOS ($t = -2.94$, 43.39, $p < 0.01$) and a 3.87 shorter DHD than the AUDm ($t(59.31) = 2.12$, $p < 0.05$); other measures did not differ significantly. However, as described in the Methods, all gender interactions with drinking history remained significant in a subsample for which the AUDw and AUDm did not differ significantly by LOS or DHD.

### Hippocampus volumes, group, age, and gender

Here we report the results for the whole hippocampus, the subfield main model ANOVA, estimated marginal mean differences, comparisons of age slopes, and then for gender effects. All volumes were adjusted for head size (eTIV) before the analyses. Linear regression analyses of whole hippocampus volume revealed a significant interaction of group-by-age ($F(1, 123) = 5.81$, $p < 0.05$); the AUD group had 5.21% smaller volumes than the NC group ($t(123) = -4.11$, $p < 0.001$). Age was associated with a 0.45 percent/year decline for the AUD group, which was significantly steeper than the 0.19 percent/year decline observed for the NC group.

To analyze the subfield volumes, we used mixed-model regression analyses, followed by post hoc analyses of estimated marginal means and post hoc analyses of slopes. As described in the Methods, we examined higher-order interactions to test for homogeneity of regression slopes, and then eliminated the interactions that were not significant. These analyses revealed a significant three-way interaction for group-by-region-by-age, as detailed in S2 Table. No significant group-by-gender interactions were observed. Group differences in means, and in slopes (for age), are reported below.

The post hoc analyses of mean differences between groups in subfield volumes (Bonferroni adjustment of family-wise $p < 0.05$ for 12 regions: $p < 0.0042$) revealed that compared to the NC group, the AUD group had significantly smaller subiculum, CA1, molecular layer, and hippocampal tail regions (Fig 2 and S2 Table). Specifically, in the AUD group, the volumes of the subiculum, CA1, molecular layer, and hippocampal tail were smaller by 4.11%, 5.16%, 5.21%, and 5.28%, respectively.

The impact of age on subfield volumes interacted significantly with group (the group-by-region-by-age interaction). The post hoc analyses showed that for the AUD group, increased

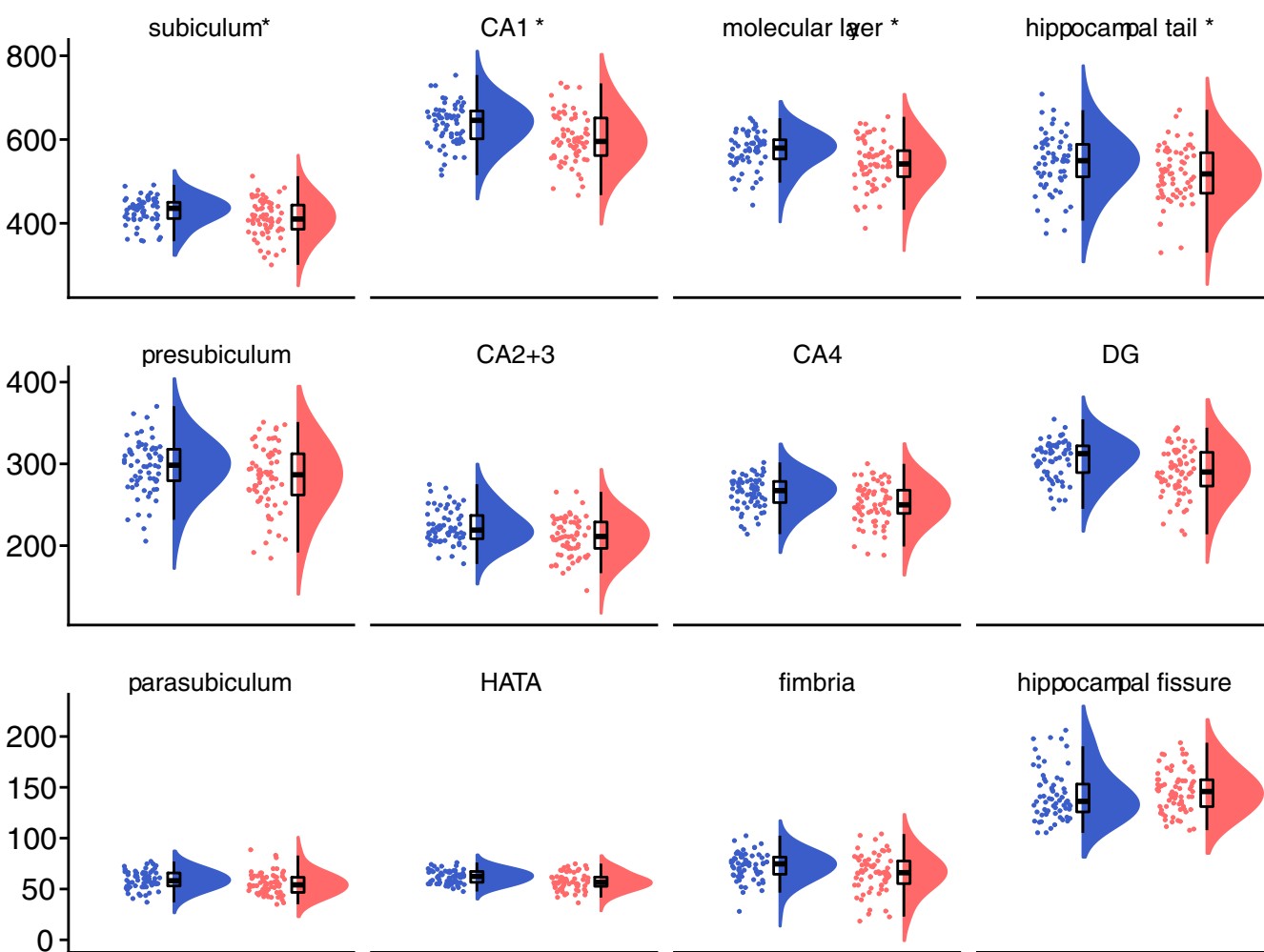

**Fig 2. Regional volumes adjusted for estimated total intracranial volume (eTIV).** Half violin raincloud plots [52] show hippocampal subfield volumes for the NC and AUD groups. See S1 Table for mean and standard deviation values. Abbreviations: CA1 = cornu ammonis 1; CA2+3 = cornu ammonis 2 and 3; CA4 = cornu ammonis 4; DG = dentate gyrus; HATA = hippocampal-amygdaloid transition area; Sub = subiculum; NC = Nonalcoholic Control group; AUD = Group with history of Alcohol Use Disorder. *Indicates regions where AUD < NC, $p < 0.0042$.

age was associated with significantly smaller volumes of the subiculum, molecular layer, and hippocampal tail (-0.44, -0.49, and -0.51, respectively; all percent/year; $p$s $< 0.0042$). These relationships were significantly more negative than those observed for the NC group (-0.10, -0.24, and -0.15; all percent/year; $p$s $< 0.0042$).

While a significant gender-by-region interaction was observed (S2 Table), post hoc comparisons between men and women in subfield volumes were not significant after correction for multiple comparisons ($p$s $> 0.0042$).

## Subfield volumes and memory

Following our main analyses, we assessed the relationship of memory scores and subfield volumes. All our secondary analyses were built upon our primary model (see Methods). We

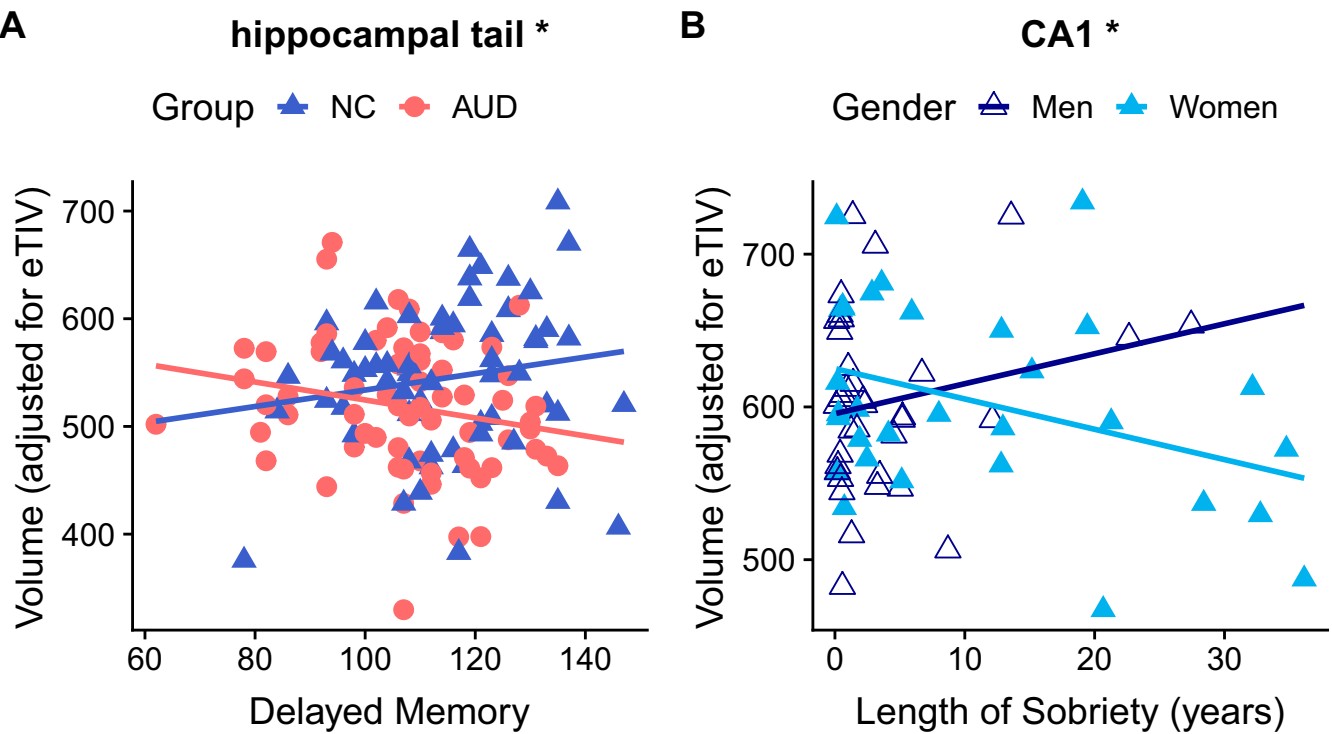

**Fig 3. Significant interactions for group-by-region-by-DMI and gender-by-region-by-LOS.** A. For the AUD group, Delayed Memory was associated with lower hippocampal tail volumes, while for the NC group, a positive relationship was observed. B. For AUD men, CA1 volumes (adjusted for eTIV) were positively associated with Length of Sobriety (years), while for AUD women, a negative relationship was observed. Abbreviations: DMI = Delayed Memory Index; LOS = Length of Sobriety; CA1 = cornu ammonis 1; NC = Nonalcoholic Control group; AUD = Group with history of Alcohol Use Disorder; eTIV = estimated total intracranial volume. * Indicates regions with interactions significant at $p < 0.0042$.

found significant interactions (Fig 3, S1 and S2 Figs, S3 and S4 Tables) for group-by-region-by-IMI and for group-by-region-by-DMI. For better interpretability, slopes were divided by the grand mean for each region and are thus presented as percent volume/IMI or DMI unit.

**Immediate memory index.** Following the identification of significant group effects for subiculum, CA1, molecular layer, and hippocampal tail, we found that associations of IMI (S3 Table) with those regions differed for AUD and NC groups in the hippocampal tail ($t(655.03) =$ -3.89, $p < 0.001$). In AUD participants, the volumes decreased by 0.13% per unit of IMI, while for NC they increased by 0.14% with each unit of IMI.

**Delayed memory index.** Also, in the hippocampal tail, a significant interaction of AUD and NC participants was observed between DMI and subfield regions (S4 Table and Fig 3A). In AUD participants, the volumes were 0.15% lower with each unit of DMI, while for NC participants, volumes were 0.22% higher with each unit of DMI, relationships that differed significantly from each other ($t(661.54) = -5.13$, $p < 0.001$).

### Subfield volumes and drinking history

For the AUD group, we examined the relationships of DHD, DD, and LOS to subfield volumes, and the interactions of those drinking history measures with gender and region. All models included age as a covariate, and correlations between age, DHD, DD, and LOS were low (all r < 0.5; age with DHD = 0.36; age with DD = -0.37; age with LOS = 0.45; DHD with DD = 0.13; DHD with LOS = -0.26; DD with LOS = -0.28). S5 Table shows that the LOS-by-gender-by-region interaction was significant (F(11, 627) = 2.34, $p < 0.01$). LOS was associated

with smaller volumes in women and larger volumes in men in the CA1 ($t$(277.25) = 2.86, $p$ = 0.0046). CA1 volumes increased by 0.38% per year of sobriety in AUDm, and decreased by 0.19% per year of sobriety in AUDw (Fig 3B).

## Discussion

Results of this study confirmed findings of smaller volumes of hippocampal subfields in association with AUD [19–21]. Compared to the NC group, the AUD participants exhibited significantly smaller volumes in the CA1, subiculum (with marginal significance), molecular layer, and hippocampal tail. Smaller volumes in these regions could result in abnormal neural processing, coincident with impairments of distinct mental functions. We also found significant associations between the volumes of individual subfields with age, memory, gender, and measures of drinking history.

Alterations in hippocampal subfield volumes have implications both for downstream targets and for hippocampal processing of upstream inputs. Accordingly, fewer neurons in the subiculum and CA1 may result in weaker output from the hippocampus, and smaller dendritic arborizations in the molecular layer may cause the hippocampus (particularly anterior regions) to be less effective in distinguishing between distinct contextual inputs, thereby resulting in forms of overgeneralization [53]. Both types of abnormality (input and output) could have severe consequences for emotion and motivation or sensitivity to reward. For example, the hippocampus projects to medial prefrontal areas, which are involved in conflict detection [54]. Weak or insufficiently precise contextual signals from hippocampus may bias these cortices towards greater sensitivity to emotional signals, such as those received from the amygdala. Elevated or overgeneralized processing of conflict may, in turn, bias the motivational and reward systems, particularly the nucleus accumbens, towards the use of poor coping strategies, including excessive drinking and other forms of self-medication. Abnormal inputs also can bias processing. For example, since the hippocampus receives connections from the amygdala [55], those signals could modulate the hippocampus more strongly if it has fewer neurons or smaller dendritic arborizations. This then could act synergistically with weak contextual representations, thereby reinforcing their emotional valence. In other words, overgeneralized contextual representations might be more susceptible to "somatic markers" [56], as well as other forms of associative learning [53], which would increase the vulnerability to contextual signals that motivate drinking.

### Subiculum and CA1

Both output regions of the hippocampus (subiculum and CA1) were reduced in volume, which could be related to two cognitive corollaries: (1) an abnormal ability to distinguish a currently experienced context from other similar previously learned contexts [57,58], and (2) higher susceptibility to emotion-driven actions [59]. That is, following a history of AUD, a given experienced context might be more likely to match a previous context in which drinking occurred.

The role of context in triggering alcohol craving is well established, and context has behavioral and treatment implications [60,61]. Evidence suggests that social support may be helpful for circumventing specific contexts entirely [62], which would avoid the aforementioned overgeneralizing activity of an impaired subiculum and CA1. The subiculum in particular has been implicated in the prediction of future rewards [63], perhaps due to its projections (along with projections of CA1) to the nucleus accumbens and ventral tegmental area. Likewise, the CA1 has been shown to play a role in mental processes involving envisioning the self in the future and past [15]. Therefore, the abnormalities we observed in these structures in association with

AUD could negatively impact motivation through the impairment of both future processing and reward prediction.

## Molecular layer

The molecular layer also was smaller in the AUD group. This region includes the apical dendrites of hippocampal pyramidal cells, and is traditionally viewed as an input region to the subfields of the hippocampus [64]. Smaller volumes of the molecular layer could suggest a relationship of AUD with hippocampal inputs or with local computational processes of the hippocampus (as opposed to connectivity exiting the structure) and could lead to outcomes that are similar to those resulting from smaller subiculum and CA1 volumes. In the molecular layer, entorhinal cortex inputs are thought to convey information from the current sensory context, while intra-hippocampal projections to the molecular layer are thought to represent a memory representation of the current context [65,66]. With reduced volume of this circuitry in the molecular layer, the quality of the processing at the meeting of these two streams of input may be impacted. Therefore, it could result in a reduced ability to compare previously experienced contexts with the current experience, and might lead the hippocampus to over-generalize the present context to previously experienced high-valence contexts. In turn, this could bias the system toward addictive behaviors, i.e., similar outcomes to those discussed regarding reductions in subiculum and CA1 volumes.

## Hippocampal tail

While the anterior hippocampus has been associated with stress, emotion, and affect [17], reductions in cells and cell processes in the posterior tail could involve many elements of spatial processing. This possibility is congruent with previous findings of alcohol-mediated alterations of spatial processing [16,67,68], which in rats, has been hypothesized to occur through alterations of hippocampal place cells [69]. In our AUD group, hippocampal tail volume was negatively associated with both immediate and delayed memory scores, in contrast to the positive association observed in the NC group. One possible interpretation for this finding involves regional compensation within the brain [16,70,71]. That is, for AUD participants with smaller hippocampal tail volumes, the extent of structural impairment might be sufficient to necessitate a shift to using other structures for the same function. Once this shift occurs, the new structures may provide good compensation for learning—but this shift may occur more often for the more extreme cases of hippocampal perturbation. In contrast, for AUD participants with mild hippocampal impairments, the brain may continue to rely on the impaired structures instead of shifting to an alternative compensatory region; relying on the impaired structures could result in impaired memory performance. (A metaphor that may assist in understanding is as follows: One may continue using a somewhat functioning toaster and make low quality toast; with a broken toaster one may switch to the oven broiler and make better toast.) Taken together, this explanation would fit with the observed tendency for larger hippocampal volumes to be associated with worse performance in the AUD group, and is consistent with findings that report how medial prefrontal cortex may compensate by increasing its efficiency for learning and memory after substantial hippocampal dysfunction [72].

## Influence of participant characteristics

While our results of smaller volumes of the subiculum, CA1, and molecular layer in AUD subjects are in agreement with past findings [19–21], our results did not indicate significant reductions in other regions that were reported in those studies, including CA2+3, CA4, HATA, and fimbria. However, abstinence durations for the AUD participants examined in those papers

generally were considerably shorter than the average abstinence durations of the participants in the present study (7.1 years). Therefore, results reported in the earlier studies could reflect effects associated with early sobriety. Moreover, those studies [19–21] did not examine the *relationship* of long durations of abstinence to volumes. In the few studies that examined total hippocampal volume 'recovery', the average length of abstinence was short, varying from a few weeks [73,74] to approximately two years [75]. One study examined a sample of AUD individuals with a history of comorbid psychiatric illnesses and an average abstinence of six years, and the authors found smaller total hippocampal volumes compared to NC individuals [76]. However, the authors reported only minimal (nonsignificant) differences in hippocampal volumes for abstinent AUD participants without the psychiatric comorbidities. In our sample of AUD participants (many of whom had psychiatric histories as shown in S6 Table), it could be that other effects of alcohol consumption were no longer evident after such long periods of abstinence. For the abnormalities in regions found to be in agreement across previous studies and ours (CA1, molecular layer, and subiculum), the combined results support the view that they are vulnerable to chronic long-term AUD.

Associations between age and hippocampal volumes were more negative in our AUD group than in our NC group for the subiculum, molecular layer, and hippocampal tail regions. These findings extend earlier reports, based on pathology and MRI investigations, that brain regions, including the hippocampus, show greater volumetric reductions or abnormal blood flow in older than in younger individuals with AUD [35,41]. Moreover, cognitive ramifications of an interaction of age and AUD were reported to exert compounded abnormalities in memory and visuospatial abilities [77]. Although in the present study, age was negatively associated with volumes of the subiculum, molecular layer, and hippocampal tail regions, Zahr and colleagues [19] reported accelerated aging in the CA2+3 subfield of AUD participants. While not resolved, it should be noted that there is a long-running hypothesis discussing how age may be associated with greater volume losses and functional decline in AUD groups than NC groups [32,46].

Gender differences also contribute to divergent findings across studies. Although the present study and the results reported by Agartz et al. [37] and Zahr et al. [19] did not show significant interactions between gender and diagnostic group, gender differences in volumes may be elucidated further by considering quantity and duration of alcohol consumption, and LOS. In the present study, CA1 volume was related to LOS differently for men and for women. In men, CA1 volume increased with LOS, suggesting recovery over time. However, in women, the volumes continued to decline with sobriety, even after statistically accounting for age. This might indicate an AUD-related abnormality in another biological system, as AUDw may be more susceptible to liver injury and heart disease, and have been reported to display lower drinking thresholds for systemic damage [78].

## Limitations

In the present study, the automated subfield labeling procedure that we employed relied upon a probabilistic atlas, rather than borders defined by image contrast at high resolution. Further, the accuracy of our automated segmentation was limited, because T2 scans were unavailable. The segmentation procedure we used (developed by Iglesias et al. [6]) generates better results when using both T2 and T1 than when using T1 alone, and our use of T1 scans without T2 scans could have resulted in lower accuracy.

We did not consider several other factors that could influence our findings (see Oscar-Berman et al. [16] for review). For example, we used a cross-sectional design, whereas a longitudinal cohort may be better able to show how the observed abnormalities were related to pre-

existing risk factors for AUD or consequences of AUD. Relatedly, our sample had a heterogeneous LOS, which prevented us from specifying time points corresponding to the percentage of inferred changes we reported in subfield volumes. Additionally, we did not consider participant characteristics (S6 Table) such as family history of AUD [79] or smoking [80,81]. We also limited the scope of this study with regard to the number of cognitive assessments we considered in our analyses, i.e., Full Scale IQ [44] and two inclusive measures of memory, i.e., immediate and delayed memory [44], that we had hypothesized to be positively associated with volumetric measures. Other WAIS and WMS measures can be found in the available data and code: https://gitlab.com/kslays/moblab-hippocampus.

As described in the Methods, although we included participants with confounding factors in order to increase the generalizability of the findings to AUD individuals in the United States population, we recognize that these confounding factors could limit interpretability. For that reason, we analyzed the data from a subsample of AUD participants without the confounding factors. All statistical group effects reported in ANOVAs, including group interactions, remained significant for this unconfounded subsample. As a separate issue, the AUDw group had longer LOS and shorter DHD than the AUDm group. As described in the Methods, we analyzed a subsample of AUDm and AUDw who did not differ significantly on LOS or DHD, and the significant gender interaction with LOS remained significant. Additionally, in the four models reported, measures of education and IQ were not included in the statistical models presented. However, we re-ran the models controlling for age and education, and all statistical group effects reported in ANOVAs, including group interactions, remained significant.

## Conclusions

The results indicated smaller input (molecular layer) and output (CA1) volumes for the AUD group, abnormalities that could be related to distorted context processing in the AUD group. Smaller volumes also were evident for the hippocampal tail, implicating deficits in spatial processing. Memory scores were negatively associated with hippocampal tail volume in the AUD group, while a positive association was observed for the NC group, suggesting that the larger volumes were associated with better performance. This finding might further signal an AUD-related functional deficit in the hippocampal tail, and the spatial processing and memory functions performed by that region. Our observation of more extreme age-related hippocampal volume reductions in the AUD group than in the NC group, not only are congruent with the notion of synergistic negative impacts of alcohol exposure and aging; they also refine the sub-regional implications of the abnormality to the subiculum, molecular layer, and hippocampal tail. Regarding gender differences, longer LOS was associated with larger CA1 volumes in AUDm, possibly indicative of recovery of contextual processing over time; however, the smaller volumes observed in AUDw in conjunction with longer sobriety periods, might suggest impaired recovery for women, perhaps tied to abnormalities in other biological systems. We believe that our findings not only build upon other work that highlights brain structural and functional abnormalities in the impact of AUD, they also suggest that clinicians, educators, and public health officials could benefit by approaching prevention and treatment strategies with respect to individual differences.

## Supporting information

**S1 Table. Regional volumes adjusted for estimated total intracranial volume (eTIV).** Means and standard deviations (SD) are provided for the hippocampal regional volumes of AUDw (N = 31) and AUDm (N = 36) (women and men with a history of Alcohol Use Disorder), along with NCw (N = 30) and NCm (N = 33) (women and men without a history of

AUD). Abbreviations: CA1 through 4 = cornu ammonis 1 through 4; DG = dentate gyrus; HATA = hippocampal-amygdaloid transition area. *Indicates regions where AUD < NC, $p < 0.0042$.
(DOCX)

**S2 Table. Analysis of variance for the primary model of our study.** The analysis of variance obtained from the model indicated significant group-by-region-by-age and gender-by-region-by-age interactions for volumes. Colons indicate interaction effects. Abbreviations: Sum Sq = sums of squares; Mean Sq = mean square; NumDF = numerator degrees of freedom; DenDF = denominator degrees of freedom; Pr(>F) = probability > F (i.e., $p$ value).
(DOCX)

**S3 Table. Analysis of variance for a secondary model of our study, which includes the Immediate Memory Index.** The analysis of variance obtained from the model indicated a significant group-by-region-by-IMI interaction for volumes. Colons indicate interaction effects. Abbreviations: Sum Sq = sums of squares; Mean Sq = mean square; NumDF = numerator degrees of freedom; DenDF = denominator degrees of freedom; Pr(>F) = probability > F (i.e., $p$ value); IMI = Wechsler Memory Scale Immediate Memory Index.
(DOCX)

**S4 Table. Analysis of variance for a secondary model of our study, which includes the Delayed Memory Index.** The analysis of variance obtained from the model indicated a significant group-by-region-by-DMI interaction for volumes. Colons indicate interaction effects. Abbreviations: Sum Sq = sums of squares; Mean Sq = mean square; NumDF = numerator degrees of freedom; DenDF = denominator degrees of freedom; Pr(>F) = probability > F (i.e., $p$ value); DMI = Wechsler Memory Scale Delayed Memory Index.
(DOCX)

**S5 Table. Analysis of variance for a secondary model of our study, which includes the AUD groups' drinking history (DHD, DD, and LOS).** The analysis of variance obtained from the model indicated a significant gender-by-region-by-LOS interaction for volumes, for the AUD group. Colons indicate interaction effects. Abbreviations: Sum Sq = sums of squares; Mean Sq = mean square; NumDF = numerator degrees of freedom; DenDF = denominator degrees of freedom; Pr(>F) = probability > F (i.e., $p$ value;.DHD = duration of heavy drinking; DD = daily drinks; LOS = length of sobriety.
(DOCX)

**S6 Table. Additional participant characteristics.** Counts of participants are given for each level of the measures listed for AUDw (N = 31) and AUDm (N = 36) (women and men with a history of Alcohol Use Disorder), along with NCw (N = 31) and NCm (N = 33) (women and men without a history of AUD). Confounded and unconfounded subsample assignment is described in the Methods. The five participants listed with Five Year Drug History of 'once per week or more' were occasional marijuana users. First Degree History was indicated by participant endorsement of 'Alcoholic' for mother, father, sibling, or children. Second Degree History was indicated by participant endorsement of 'Alcoholic' for grandparents, aunts, uncles, or grandchildren.
(DOCX)

**S1 Fig. Relationships of volume with delayed memory for AUD and NC groups.** As described in Fig 3, for the AUD group, Delayed Memory Index was associated with smaller hippocampal tail volumes (adjusted for eTIV), while for the NC group, a positive relationship was observed. This figure shows the relationships for all 12 regions. Abbreviations:

CA1 = cornu ammonis 1; CA2+3 = cornu ammonis 2 and 3; CA4 = cornu ammonis 4; DG = dentate gyrus; HATA = hippocampal-amygdaloid transition area; Sub = subiculum; eTIV = estimated total intracranial volume. *Indicates regions where $p < 0.001$ for the interaction of group-by-Delayed Memory Index.
(EPS)

**S2 Fig. Relationships of volume with length of sobriety for AUD men and women.** As described in Fig 3, for AUD men, CA1 volumes (adjusted for eTIV) were positively associated with Length of Sobriety, while for AUD women, a negative relationship was observed. This figure shows the relationships for all 12 regions. Abbreviations: CA1 = cornu ammonis 1; CA2+3 = cornu ammonis 2 and 3; CA4 = cornu ammonis 4; DG = dentate gyrus; HATA = hippocampal-amygdaloid transition area; Sub = subiculum; eTIV = estimated total intracranial volume. *Indicates regions where $p < 0.01$ for the interaction of group-by-Length of Sobriety.
(EPS)

## Acknowledgments

The authors thank Howard Cabral, Zoe Gravitz, Yohan John, Steve Lehar, Riya Luhar, Nikos Makris, Pooja Parikh Mehra, Diane Merritt, Greg Millington, Jason Tourville, Maria Valmas, and Andrew Worth for assistance with recruitment, assessment, data analyses, neuroimaging, or manuscript preparation. We also wish to acknowledge the Athinoula A. Martinos Center of Massachusetts General Hospital for imaging resources, and the Boston University Clinical and Translational Sciences Institute (BU-CTSI) for statistical consultation. We further appreciate the suggestions provided by the reviewers, especially for recommending that we implement the method for estimated total intracranial volume correction. Finally, we would like to acknowledge the role of the research participants for making this study possible.

## Author Contributions

**Conceptualization:** Kayle S. Sawyer, Susan M. Ruiz, Marlene Oscar-Berman.

**Data curation:** Kayle S. Sawyer, Noor Adra, Daniel M. Salz, Susan M. Ruiz.

**Formal analysis:** Kayle S. Sawyer, Noor Adra, Daniel M. Salz.

**Funding acquisition:** Marlene Oscar-Berman.

**Investigation:** Kayle S. Sawyer, Susan M. Ruiz.

**Methodology:** Kayle S. Sawyer, Daniel M. Salz.

**Project administration:** Kayle S. Sawyer, Maaria I. Kemppainen, Gordon J. Harris, Marlene Oscar-Berman.

**Resources:** Susan M. Ruiz, Marlene Oscar-Berman.

**Software:** Kayle S. Sawyer, Noor Adra.

**Supervision:** Kayle S. Sawyer, Daniel M. Salz, Maaria I. Kemppainen, Marlene Oscar-Berman.

**Visualization:** Kayle S. Sawyer, Noor Adra.

**Writing – original draft:** Kayle S. Sawyer, Noor Adra, Daniel M. Salz, Maaria I. Kemppainen, Susan M. Ruiz, Marlene Oscar-Berman.

**Writing – review & editing:** Kayle S. Sawyer, Noor Adra, Daniel M. Salz, Marlene Oscar-Berman.

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
