## [Decision Letter · Decision Letter 0]

9 Jan 2020

PONE-D-19-32308

Hippocampal subfield volumes in abstinent men and women with a history of alcohol use disorder

PLOS ONE

Dear Dr. Sawyer,

Thank you for submitting your manuscript to PLOS ONE. After careful consideration by 3 Reviewers and an Academic Editor, all of the critiques of all three Reviewers must be addressed in detail in a revision to determine publication status. If you are prepared to undertake the work required, I would be pleased to reconsider my decision, but revision of the original submission without directly addressing the critiques of the three Reviewers does not guarantee acceptance for publication in PLOS ONE. If the authors do not feel that the queries can be addressed, please consider submitting to another publication medium. A revised submission will be sent out for re-review. The authors are urged to have the manuscript given a hard copyedit for syntax and grammar.

**Comments to the Author**

1. Is the manuscript technically sound, and do the data support the conclusions?

Reviewer #1: Yes

Reviewer #2: Partly

Reviewer #3: Yes

2. Has the statistical analysis been performed appropriately and rigorously? 

Reviewer #1: Yes

Reviewer #2: Yes

Reviewer #3: I Don't Know

3. Have the authors made all data underlying the findings in their manuscript fully available?

Reviewer #1: Yes

Reviewer #2: Yes

Reviewer #3: Yes

4. Is the manuscript presented in an intelligible fashion and written in standard English?

Reviewer #1: Yes

Reviewer #2: Yes

Reviewer #3: Yes

5. Review Comments to the Author

Reviewer #1: The manuscript describes an analysis of the hippocampal subfields volumes in a large sample of AUD patients compared with controls. The authors also examines the relationships with general memory scores and alcohol history. The manuscript is well-written and easy to follow despite the numerous analyses. I particularly appreciate the strategy for the statistical analyses: large samples representative of the clinical population and smaller ones much more carefully selected to avoid the effects of confounding factors. The statistical analyses are rigorously chosen. The results are sometimes surprising and counterintuitive but the authors manage to discuss them quite well.

There is no page number, neither line number, which makes it difficult to refer to.

Introduction

“negative association between… (clinical vs subclinical)”: could you detail?

“This processing stream is repeated… ventral to dorsal”: could you specify? Which processing stream? What is organized from anterior to posterior? Do you refer to the nature of the stimuli that are processed? Or to the nature of the process per se?

“the problems contributing to, and resulting from, AUD”: the fact that altered memory abilities result from AUD makes sense. Can you explain how shrinkage of specific hippocampal subfield and altered memory abilities contribute to AUD? You explain this point only in the discussion.

Methods

Why did the inclusion criteria require a minimum of 4 weeks of abstinence? The withdrawal syndrome is over well before 4 weeks.

Why did the authors focus on these two memory indexes? What about the three other ones? Do they have specific hypotheses with these ones?

What are the “composite scores” of the WAIS? Not mentioned in Table 1 (just full IQ).

Since the authors explain that the different hippocampal subfields have different cognitive functions, why did they use a very global memory scale, which does not permit to evaluate these specific cognitive/memory processes?

Discussion

Previous studies reporting shrinkage of other hippocampal regions included patients with shorter LOS. I do not think that with an average of 7 years of sobriety, results can reflect acute effects of alcohol.

Tables/Figures

Table 1 is difficult to read because of too many abbreviations. Please, include the statistics in the table to facilitate the reading.

LOS is supposed to be in years given the legend of Table 1. Have the women AUD really been sober in average for 11 years (with a maximum of 36 years)? If so, the sample (even in men AUD) is very heterogeneous regarding the LOS, which is nice to evaluate recovery effect but makes it difficult to state on the presence/absence of alterations at a specific time point. For example, the 5% smaller volumes of different hippocampal fields corresponds to the volumes at 4 weeks (certainly not), 6 months, 6 years of sobriety?

I cannot find the * in Figure 2 (as indicated in the legend).

Figure 3: B appears before A in the text.

Reviewer #2: This paper looks at hippocampal subfield volumes in AUD and tries to elucidate age-sex-group relationships. This is a fairly new area with few prior work, aided by the recent availability of hippocampal subfield segmentation software (Freesurfer).

General comments

There are two major limitations of the paper which could potentially have contributed to their results being different from prev. work and calling into question some of the gender effects claimed here-

1. The MPRAGE T1 have poor slice resolution (1.5mm) but more importantly only T1 data was used for segmentation. Freesurfer 6.0 has been shown to be improved over 5.x only because of the ability to jointly analyze T1 and T2 data. This is clearly the motivation of Iglesias et al. when they developed FS6.0. However, in this work only T1 was used which puts into question the validity of volumes measures of small structures like the molecular layer which are hardly even visualized in T1. It is likely this cannot be corrected but if the authors have access to the T2 data, they should use it in the FS pipeline to significantly improve accuracy. Otherwise, they cannot claim the superior segmentation of 6.0 over 5.X.

2. The other major and perhaps critical limitation (which can be fixed !) is the correction for ICV. It has been shown in several papers (Pintzka et al. Front Neurosci. 2015; 9: 238. Voevodskaya Front. Aging Neurosci., 07 October 2014 ) that the method of ICV correction is critical esp. in determining sex correlations to volumes. It has been shown either the residual or the ANCOVA method are the most accurate and the proportions based method (i.e. vol/ICV) the least accurate and leading to confounding results. Since one of the main claims of this paper is sex specific effects, this has to be addressed or the results are questionable. The residual method is very easy to implement (fit the control volumes to the ICVs, use the slope s to correct AUD vols as V' = V - s*(ICV- mean ICV).

Specific comments

Introduction: It is possible that a correct and accurate ICV correction will eliminate the gender effect and possibly the reason why there were no prev. gender specific effects reported in prev. studies. [See major comment above].

The introduction is a bit too long and rambling and didactic. Anatomy of the hippocampal subfields etc can be removed and summarized in the Discussion.

Abstract: "higher scores" please clarify what these scores are

Results: "AUD group had 5.18%, 5.08%..." -- if there are sex effects, shouldn't this be separated by sex? Similarly in the next para, it says Women had 8.43%, 4.99% etc but this should also be separated by group to see what those effects are (if there are) and then combined if there are indeed no effects.

For the age effects, it suddenly switches to %/year. It is not clear how this was calculated. Was it just based on the age spread? I assume there were no multiple time point scans. So this is a bit confusing why units changed from % to % per year.

Again, the accuracy of women having 0.4%/year reduction in CA1 volumes vs. men is questionable based on the way the ICV normaization was done. Same for most other results involving sex differences.

Reviewer #3: Hippocampal Subfield Volumes in Abstinent Men and Women With a History of Alcohol Use Disorder

Manuscript Number: PONE-D-19-32308

This manuscript examines volumes of hippocampal subfields in a sample of men and women with and without a history of alcohol use disorders. Analyses find several group differences in hippocampal subfield volumes, with main effects of group moderated by interactions with gender, age, and drinking phenotypes. The manuscript is very well written, the rationale is clearly laid out in the Introduction, relevant literature and background is reviewed, and the Discussion is well structured. However, I had some trouble following along with the analyses and results, particularly the multiple 3-way (and even a few 4-way) interactions. I make a few suggestions for this point and have a few additional points below that I hope might improve what is already a strong manuscript.

1. Please indicate in the Abstract and description of the sample how long participants in the alcohol use disorder group have been abstinent. Length of sobriety is noted in the text/tables—is this the most recent period of sobriety, or the longest length of sobriety in the lifetime?

2. Regarding abstinence, how should these results in this sample be interpreted in light of evidence from human and animal studies of hippocampal volume “recovery” following alcohol abstinence? Could it be that some effects of alcohol on the hippocampus are no longer evident in this abstinent sample?

3. I had some trouble following the analyses and results. Were the two 3-way interactions of group x gender x region and group x gender x age both included in a single model, as suggested by Table S2? Why was group x gender x memory (or group x age x memory?) not examined, but instead group x region x memory was, as in Tables S3 and S4?

4. Related to the above point, I had some trouble following the results for the different 3-way interactions. It seemed that results for main effects were reported (e.g., the third paragraph in the Hippocampal Volumes, Group, Gender, and Age section describes group-level mean differences, but shouldn’t this paragraph be unpacking the 3-way group x region x age model described in the previous paragraph?

5. I appreciate the efforts to minimize false positive findings, but it wasn’t clear to me why the Bonferroni correction was set as .050/12 (the number of hippocampal subfields) for some analyses but not others. Far more than 12 analyses were conducted, and 12 seems like a fairly arbitrary number. I’m not necessarily suggesting a more stringent p value, but perhaps some additional rationale would be helpful here.

6. PLOS authors have the option to publish the peer review history of their article (what does this mean?). If published, this will include your full peer review and any attached files.

**Do you want your identity to be public for this peer review?** For information about this choice, including consent withdrawal, please see our Privacy Policy.

Reviewer #1: Yes: Anne Lise PITEL

Reviewer #2: No

Reviewer #3: No

We would appreciate receiving your revised manuscript by July, 2020. To enhance the reproducibility of your results, we recommend that if applicable you deposit your laboratory protocols in protocols.io, where a protocol can be assigned its own identifier (DOI) such that it can be cited independently in the future. For instructions see: http://journals.plos.org/plosone/s/submission-guidelines#loc-laboratory-protocols

We look forward to receiving your revised manuscript.

Kind regards,

Stephen D. Ginsberg, Ph.D.

Section Editor

PLOS ONE

"This work was supported by funds from the National Institute on Alcohol Abuse and Alcoholism (NIAAA; https://www.niaaa.nih.gov/) of the National Institutes of Health US Department of Health and Human Services under Award Numbers R01AA07112 and K05AA00219 awarded to M.O.B.; US Department of Veterans Affairs Clinical Science Research and Development (https://www.research.va.gov/services/csrd/) grant I01CX000326 awarded to M.O.B. The funders had no role in study design, data collection and analysis, decision to publish, or preparation of the manuscript."

We note that one or more of the authors are employed by a commercial company: Sawyer Scientific, LLC.

"This work was supported by funds from the National Institute on Alcohol Abuse and

Alcoholism (NIAAA) of the National Institutes of Health US Department of Health and Human

Services under Award Numbers R01AA07112 and K05AA00219; US Department of Veterans

Affairs Clinical Science Research and Development grant I01CX000326; and shared

instrumentation grants 1S10RR023401, 1S10RR019307, and 1S10RR023043 from the National

Center for Research Resources (now National Center for Advancing Translational Sciences) at

the Athinoula A. Martinos Center, Massachusetts General Hospital, and Boston University

Clinical and Translational Sciences Institute (BU CTSI; 1UL1TR001430)."

"This work was supported by funds from the National Institute on Alcohol Abuse and Alcoholism (NIAAA; https://www.niaaa.nih.gov/) of the National Institutes of Health US Department of Health and Human Services under Award Numbers R01AA07112 and K05AA00219 awarded to M.O.B.; US Department of Veterans Affairs Clinical Science Research and Development (https://www.research.va.gov/services/csrd/) grant I01CX000326 awarded to M.O.B. The funders had no role in study design, data collection and analysis, decision to publish, or preparation of the manuscript."

---

## [Author Response · Author response to Decision Letter 0]

29 May 2020

Please see the attached response to reviewers.

---

## [Decision Letter · Decision Letter 1]

24 Jun 2020

PONE-D-19-32308R1

Hippocampal subfield volumes in abstinent men and women with a history of alcohol use disorder

PLOS ONE

Dear Dr. Sawyer,

Thank you for resubmitting your work to PLOS ONE. Please make the corrections posed by Reviewer #2 so I can render a decision on this manuscript.

**Comments to the Author**

1. If the authors have adequately addressed your comments raised in a previous round of review and you feel that this manuscript is now acceptable for publication, you may indicate that here to bypass the “Comments to the Author” section, enter your conflict of interest statement in the “Confidential to Editor” section, and submit your "Accept" recommendation.

Reviewer #2: All comments have been addressed

Reviewer #3: All comments have been addressed

2. Is the manuscript technically sound, and do the data support the conclusions?

Reviewer #2: Yes

Reviewer #3: Yes

3. Has the statistical analysis been performed appropriately and rigorously? 

Reviewer #2: Yes

Reviewer #3: Yes

4. Have the authors made all data underlying the findings in their manuscript fully available?

Reviewer #2: Yes

Reviewer #3: Yes

5. Is the manuscript presented in an intelligible fashion and written in standard English?

Reviewer #2: Yes

Reviewer #3: Yes

6. Review Comments to the Author

Reviewer #2: The authors have done a good job of addressing most of my comments in the first review. I still have a few minor comments--

1. We wish to note that Iglesias and colleagues had built the segmentation classifier used in

the FS 6.0 subroutine with T1-only segmentation in mind, as described in their paper as follows:

“The use of a simple, linear classifier such as LDA ensures that the classification accuracy is

mainly determined by the quality of the input data (i.e., the subregion volumes) rather than

stochastic variations in the classifier.”

The line quoted from the paper does not imply in any way FS6 had only T1 segmentation in mind. In fact, contrary to that, the paper shows better performance when using both T2 and T1 and the whole extra contrast high in-plane resolution image as an additional input was developed to address the limitations of T1. Please remove this from Discussion or rewrite to reflect what Iglesias et al. have stated. "However, Iglesias and colleagues had built the

589 segmentation classifier used in the FS 6.0 subroutine with T1-only segmentation in mind (see

590 page 133 of their paper)."

2. "We found no gender-by-region-by-age interaction, nor any gender effects for individual

subfields after correction for multiple comparisons. We have updated all the results in the text,

and therefore, removed the gender specific results."

Can the abstract be updated to incorporate this as one of the goals is to demonstrate gender specific effects which seem muted after the residual method? I also see no change in Discussion despite a lot of gender specific stuff taken out in the new version. Please also check Discussion to update based on the new results.

3. The introduction is very rambling and can be easily shortened without losing its message. But more importantly, the final paragraph does not succinctly say what the authors set out to do. Instead it editorializes like "Ultimately, we hope that the findings will help lead to an increased understanding of how subregional hippocampal shrinkage contributes to altered memory abilities in AUD men and women as they age". This belongs in Discussion not in Intro. The first paragraph of methods (We examined.....) might make more sense at the end of the Introduction.

4. Statistical analyses- "Because brain volumes vary with head size, we used normalized volume values"-- You are not normalizing but rather correcting for ICV. Perhaps a carryover from the proportions method? Also please clarify that mean eTIV (and slope) was over the controls but the correction was applied to both groups just to be precise.

Reviewer #3: Hippocampal Subfield Volumes in Abstinent Men and Women With a History of Alcohol Use Disorder

Manuscript Number: PONE-D-19-32308_R1

This revised manuscript examines volumes of hippocampal subfields in a sample of men and women with and without a history of alcohol use disorders. Analyses find several group differences in hippocampal subfield volumes, with main effects of group moderated by interactions with gender, age, and drinking phenotypes. I was Reviewer 3 on the original version of the manuscript. As for the original manuscript, the revised manuscript is very well written, the rationale is clearly laid out in the Introduction, relevant literature and background is reviewed, and the Discussion is well structured. The authors have been very responsive to my own comments/suggestions and to the other Reviewers. I particularly appreciate their transparency in posting data and code and revised analyses. The revised manuscript is a strong one with important implications for the field of alcohol research.

7. PLOS authors have the option to publish the peer review history of their article (what does this mean?). If published, this will include your full peer review and any attached files.

**Do you want your identity to be public for this peer review?** For information about this choice, including consent withdrawal, please see our Privacy Policy.

Reviewer #2: No

Reviewer #3: No

We look forward to receiving your revised manuscript.

Kind regards,

Stephen D. Ginsberg, Ph.D.

Section Editor

PLOS ONE

---

## [Author Response · Author response to Decision Letter 1]

7 Jul 2020

Please see the attached response to the reviewers.

---

## [Editor Report · Decision Letter 2]

13 Jul 2020

Hippocampal subfield volumes in abstinent men and women with a history of alcohol use disorder

PONE-D-19-32308R2

Dear Dr. Sawyer,

We’re pleased to inform you that your manuscript has been judged scientifically suitable for publication and will be formally accepted for publication once it meets all outstanding technical requirements.

Kind regards,

Stephen D. Ginsberg, Ph.D.

Section Editor

PLOS ONE

---

## [Editor Report · Acceptance letter]

23 Jul 2020

PONE-D-19-32308R2 

Hippocampal subfield volumes in abstinent men and women with a history of alcohol use disorder 

Dear Dr. Sawyer:

I'm pleased to inform you that your manuscript has been deemed suitable for publication in PLOS ONE. Congratulations! Your manuscript is now with our production department. 

Kind regards, 

on behalf of

Dr. Stephen D. Ginsberg 

Section Editor

PLOS ONE